# Psychosocial Impact of Huntington’s Disease and Incentives to Improve Care for Affected Families in the Underserved Region of the Slovak Republic

**DOI:** 10.3390/jpm12121941

**Published:** 2022-11-22

**Authors:** Katarína Hubčíková, Tomáš Rakús, Alžbeta Mühlbäck, Ján Benetin, Lucia Bruncvik, Zuzana Petrášová, Jitka Bušková, Martin Brunovský

**Affiliations:** 1Neuropsychiatric Department, Psychiatric Hospital of Philipp Pinel in Pezinok, 90201 Pezinok, Slovakia; 2Third Faculty of Medicine, Charles University in Prague, 10000 Prague, Czech Republic; 3Department of Psychiatry, Slovac Medical University, 83303 Bratislava, Slovakia; 4Department of Neuropsychiatry, kbo-Isar-Amper-Klinikum, 84416 Taufkirchen (Vils), Germany; 5Department of Neurology and Center of Clinical Neuroscience, 1st Faculty of Medicine, Charles University and General University Hospital in Prague, 12821 Prague, Czech Republic; 6Department of Neurology, University Hospital of Ulm, 89081 Ulm, Germany; 7Landesklinikum Hainburg, 2410 Hainburg an der Donau, Austria; 8National Institute of Mental Health, 25067 Klecany, Czech Republic

**Keywords:** Huntington’s disease, psychosocial effects, underserved region, disease burden, genetic counselling, genetic testing

## Abstract

Introduction: Huntington’s disease (HD) is often on the margin of standard medical practice due to its low prevalence, the lack of causal treatment, and the typically long premanifest window prior to the onset of the symptoms, which contrasts with the long-lasting burden that the disease causes in affected families. Methods: To capture these socio-psychological aspects of HD and map the experiences of affected individuals, persons at risk of HD, and caregivers, we created a questionnaire using a qualitative research approach. The questionnaire containing 16 questions was conducted online for a period of three months through patient associations in Slovakia and their infrastructures. Results: In total, we received 30 responses. The survey results, in particular, indicate insufficient counselling by physicians with explicitly missing information about the possibility of preimplantation genetic diagnostic. There was also a necessity to improve comprehensive social and health care in the later stages of the disease, raise awareness of the disease in the general health community, and provide more information on ongoing clinical trials. Conclusion: The psychosocial effects, as well as the burden, can be mitigated by comprehensive genetic counselling as well as reproductive and financial guidelines and subsequent therapeutic programs to actively support patients, caregivers, children, and adolescents growing up in affected families, preferably with the help of local HD community association. Limitations: We have used online data collection to reach a wider HD community, but at the same time, we are aware that the quality of the data we would obtain through face-to-face interviews would be considerably better. Therefore, future studies need to be conducted to obtain more detailed information.

## 1. Introduction

Huntington’s disease (HD) is an autosomal dominant, neurodegenerative disorder leading to the progressive impairment of motor skills, cognitive function, and psychiatric disorders [1]. HD is caused by the expansion mutation of the cytosine, adenine, and guanine (CAG) triplet on chromosome 4, as identified in 1983 [2]. The huntingtin gene (*HTT,* OMIM 613004), formerly IT-15, was then identified in 1993 [3] Huntington’s disease Collaborative Research. The *HTT* gene is responsible for coding the Huntingtin (HTT) protein. The expansion mutation of the *HTT* gene leads to the production of mutant HTT (mHTT) protein accumulating in the striatum of the basal ganglia, leading to progressive neuronal dysfunction and death, usually within a few decades after the onset of symptoms [4]. The number of CAG repeats in *HTT* is inversely associated with disease onset; the greater the CAG number, the earlier the onset of HD [5]. The onset of the disease is defined as a manifestation of significant motor or neurological symptoms and occurs on average around the age of 40 [6].

The number of CAG repeats within the HTT gene varies from 6 to 35 in the general population [7]. CAG repeats in the range of 36 to 39 refer to reduced penetrance repeat length [8]. CAG repeats with less than 27 present a stable transmission, and no manifestation is expected. The CAG repeat lengths of 27 to 35 refer to the intermediate allele and are also not associated with HD development. However, there is the possibility of expansion upon transmission, resulting in genetic anticipation [9].

Especially in paternal transmission, the number of repetitions in sperm increases due to the meiotic instability of the abnormal CAG repeat, often leading to Juvenile Huntington’s Disease (JHD) with a presence of more than 60 CAG repeats and occurs before the age of 21, which comprises about 5% of all HD cases [10,11]. For this reason, the children of an affected parent (common father) may be at risk of developing the disease earlier than their parent with HD [12].

Current HD management focuses on symptomatic treatment in terms of pharmacological and non-pharmacological interventions, such as psychotherapy, physiotherapy, and speech and occupation therapies by a multidisciplinary team [13]. Several promising methods are designed to modify the natural history of HD and are currently under investigation in research studies and clinical trials aiming to reduce the reproduction of mutant *HTT* gene products [14]. The natural history of HD with the progression of symptoms in individual stages of HD about disability and life milestones, including suicidal tendencies, is illustrated in Figure 1.

Caring for patients with HD represents a gradually increasing burden for their caregivers, involving family members, partners, and other relatives lasting several decades [17]. The situation is further complicated due to the reconciliation with a 50% risk of disease in their offspring, which significantly increases emotional burden and suffering compared to other non-hereditary conditions. The genetic nature of HD means that it is not uncommon for children to care for their parents before they develop symptoms, or for a parent to care for a partner and then for the child, and sometimes for several generations to care for them at the same time. Such a chronic and widespread role of care in families with HD can deepen isolation, feelings of loss of their family members and their future, feelings of guilt, anger, burden, and helplessness [18]. The diagnosis of HD also affects the wider blood relatives.

Physicians and healthcare professionals consulting families with HD should do their utmost to ensure that all family members obtain the necessary information to make an informed decision about genetic testing, reproduction, and future financial and life plans [19,20]. Appropriate genetic counselling plays an important role in HD management. The requirements for genetic counselling differ on the condition of a client or patient, depending on the type of consultation. Two types of procedures are available: differential diagnostic and presymptomatic (predictive) testing. The predictive genetic test allows for measuring CAG repeat length in a blood laboratory sample and predicting whether an at-risk individual will develop HD later in life [19]. Predictive testing should be reserved for adults who have participated in a thorough consultation with a genetic counsellor or other trained professional or HD expert about their genetic risks and the potential risks and benefits of the test itself, as well as post-test support [21,22]. The participant needs to prepare for an unfavourable result by receiving information about the disease and the social consequences of the genetic test [22]. The most important predictive and diagnostic genetic testing procedure is appropriate counselling; informed consent from the patient must be obtained before testing [19,21]. The implications of a positive result for the patient and their family must be clearly outlined. Both the nature of HD itself and its autosomal dominant inheritance must be explained to make it clear that if the test is positive for HD, any of their children would have a 50% chance of inheriting it [20,23,24]. The usual standard process of genetic counselling and testing is summarised in Table 1.

In addition, it is appropriate to enrich the education with knowledge about the expression of HD within an intermediate allele of 27–35 CAG repeats and the impact of reduced penetration between 36–39 CAG repeats [25]. At the same time, information about technological advances in preimplantation genetic diagnostic testing (PGD) and prenatal testing improved collaborative research networks need to be discussed. A better understanding of the psychosocial aspect of living with HD needs to be provided. The HD expansion mutation carrier, manifest patients and persons at risk of HD need to be informed of their reproductive potential in preference to planned pregnancy according to the regulation of the respective countries [26]. Determining genetic status confronts people with new dilemmas in their reproductive decisions. 

It is important to highlight the HD community’s local peer support. Anyone affected by HD can benefit from participating in the activities of local HD associations, such as support group seminars on the latest information, and thus be in touch with others experiencing the same situation. In our European area is the European Huntington Association (EHA), an umbrella organisation that provides an extensive network of people coping with the disease—either in person or professionally. It accelerates the information flow and support to people with HD in their country and builds cross-border relationships [27]. The patient community comprises people personally affected by HD (with a local presence in most European countries), and the research community includes experts working with HD. The situation of affected patients and their relatives was unsatisfactory in the Slovak Republic for a long time. Although the Slovak self-help group was officially active from 1994 to 2016, it was founded and maintained by only a few volunteers in Slovakia who, despite their heroic efforts, did not receive enough support from the government and health professionals and burnt out over time. A functioning infrastructure was not developed, and even such simple things as a functioning website and communication channels with members were not established and were only based on personal contacts. This was in great contrast to the situation in the Czech Republic, with which Slovakia was a common state until 1993, so the issue of the HD community had practically the same starting point at that time. Furthermore, due to the cultural and linguistic proximity of the two countries, the affected families in Slovakia had the opportunity to compare the situation and were understandably frustrated. The Czech Huntington’s Association quickly became a member of the EHA, and at the same time, several specialised centres and/or several EHDN clinics were established in the country. Comprehensive care with options such as in vitro fertilisation (IVF), including preimplantation diagnostics, is covered by statutory health insurance for patients with Huntington’s disease. At the time of conducting this survey, there was no EHDN clinic in Slovakia and IVF, including preimplantation genetic diagnosis, has to be paid for by the patients themselves. At the end of 2019, the Slovak HD Society resumed its activities at the request of patients and their families and shortly afterwards became a member of the EHA. The support group created a dynamic exchange platform to promote and provide better access to the needs of those affected. Work began on building an infrastructure with a website. The main goal is to provide support and information about the nature of the disease, its inheritance, the possibilities of genetic diagnosis, treatment and care for patients and family members, and the establishment of support groups [28]. 

This survey was the first opportunity to officially express experiences and suggestions for improving the situation of the until then unheard HD community. It was the starting point for defining the first and next steps for the restoration of the Slovak HD association. The presented paper aims to explore and evaluate the psychosocial impacts of HD and incentives to improve the care of affected patients and families in the underserved region of Slovakia.

## 2. Material and Methods

### 2.1. Study Assessment

The project was set up to map the situation of patients and families affected by HD in the Slovak Republic (SR). This project is part of a larger national initiative to improve the care of affected patients and families. For this purpose, two types of questionnaires with 16 questions were designed to explore the main psychological aspects and domains. The questionnaire was implemented and accessible via the online platform www.survio.com accessed on 1 November 2022. The first questionnaire was aimed at people who have tested positive for Huntington’s disease, regardless of the stage of the disease. The second questionnaire focused on people at risk, not affected by HD and caregivers and partners of affected individuals. Each questionnaire consisted of 13 open questions and three multiple-choice questions with the possibility to answer in the text. The questionnaire covered the main domains of genetic counselling about Huntington’s disease knowledge (initial information, main source of information, reproductive counselling, experimental treatment, lack of knowledge), genetic testing (reasons for testing or not), the possibility of psychological counselling, the impact of Huntington’s disease (the most burdensome symptom, financial burden, impact on relationships) and social and health services in Slovakia. The questionnaire was available online for three months in the year 2020. The questionnaire was designed by an expert group of representatives of the patient HD community and health care professionals, such as the neurologist, the psychiatrist, and the psychologist and was reviewed to ensure proper understanding and clear language This survey, including a questionnaire, has been approved by a constituted Ethics Committee of the Psychiatric Hospital of Philipp Pinel in Pezinok, Slovakia (Approval Date: 30 September 2020) and conforms to the provisions of the Declaration of Helsinki in 1995 (as revised in Edinburgh in 2000).

### 2.2. Study Participants

The regular members of the Slovak HD Society—specifically patients, persons at risk and their relatives have received the possibility to participate in this project. Membership in the newly renewed Slovak HD society was obtained through online form through the newly opened website huntington.sk in the course of 2020, while information about Slovak HD society and the possibility of membership was also sent in the form of a leaflet to all genetic clinics and all neurological, psychiatric and neuropsychiatric inpatient facilities in the Slovak Republic at the beginning of 2020. As part of filling out the membership application form, it was possible, among other things, to voluntarily choose whether the applicant for membership is a patient, a person at risk, a relative, or a medical professional or write his or her other motivation to join verbally. The links to the questionnaire were sent to the regular members of the Slovak HD Patient Society, who stated that they are patients, persons at risk or relatives when applying for membership. They previously provided an email address and consent to participate in the research project to the patient’s network administrator.

In total, 60 regular members have been contacted and received an email with an explanation text and a questionnaire link. The connection to the patient questionnaire was sent on two occasions to the members to participate in the project. It was possible to fill out two questionnaires per family by repeatedly accessing the link. In the questionnaire for patients, it was possible to indicate in the opening multiple questions that the answers were recorded by another close person since the patient was no longer able to fill them in himself. Three patients used this option. The data have been proceeding anonymously, without any identifiable parameters. 

## 3. Results

In this project, the qualitative methodological approach was used to evaluate the finding of two questionnaires with 13 open-ended questions and as well the quantitative analysis to evaluate three multiple-choice questions. The survey data from the open-ended questions were analysed based on the grounded theory approach [29,30]. Repeated reading and coding of the results of the open-ended questions resulted in several themes related to the questions, and the answers were clustered into the four main domains covering: (I)Knowledge about HD, including initial information about HD, the main source of information, reproductive options, information on experimental treatment, and additional information.(II)Genetic testing and counselling, including the process of testing and psychological support and reasons why an individual decided to take a genetic test or not to be tested.(III)Impact of HD on the relationships, financial burden and burden to HD (symptoms).(IV)Social and health care services in Slovakia, including experience and proposal for improvement.

In total, 60 links to access the questionnaire were sent to the regular members of the Slovak HD Society. The responsiveness rate was 50% (30 answers). In this study, we obtained responses from the following individuals: nine individuals tested positive for HD, four of whom were asymptomatic and five of whom had the manifest disease, five individuals at risk for HD, and 16 responses from caregivers and partners of asymptomatic and manifest patients, as summarised in the first line of Table 2. 

For individual categories of questions, we also present citations of selected answers that authentically illustrate the situation and care of affected individuals with HD and its psychosocial effects. As stated above, the categories were clustered into domains I -IV to ease the overview. 

### 3.1. First Information on HD

Only three out of thirty respondents mentioned receiving a good source of initial information about the HD from their physician. However, all participants lacked information on reproductive options, instruction on preimplantation testing in in vitro fertilisation, and a generally more empathic approach. Seven participants even learned about the possibility and accessibility of the PID in Slovakia for the first time only from our questionnaire. Moreover, all respondents would better appreciate at least some information on the ongoing clinical trials. 

To the question ‘*How did your physician approach you?*’ we provide some examples: “*There was no information … I searched for everything on the internet myself*”. “*The neurologist didn’t pay attention and immediately wrote us off as an incurable diagnosis*” .“*The neurologist told me not to make a ‘big head’ out of it*”. “*I even had to actively explain to the general practitioner what the disease was and what it was causing*” “*.It was told me that I will die.*”.

We addressed the questions about the information that a person at risk for HD should receive and what to know earlier upon the genetic testing or disease onset. We received a detailed answer stating that “*greater emphasis should be placed on possible symptomatic treatment so that one does not take it for final*”. The participants would also appreciate *information on support possibilities for caregivers*, “*we have to unite to help each other; there is no other support for us*”. The participants were missing information on how to approach the affected person: “*I have often heard from a husband that I am healthy and do not understand him*”.

### 3.2. Genetic Counselling and Testing

In the section about genetic testing, we explore the reasons for testing and experience with genetic counselling. The participants stated they mainly sought this information as they needed to know the truth about planning for the future and emerging symptoms.

Of particular interest are the answers of persons at risk who either did not yet know their genetic status, as they either did not take the genetic test or were waiting for the test results at the time of participating in the survey. Some answers pointed out the option not to know about the genetic status, “We decided not to know” *on the other hand, some participants stated in more detail*, “*when planning the future, we chose the path of PGD with prenatal diagnosis, but without success. Subsequently, we decided to ignore the fear of the disease and live life as it is and as it will come. I don’t want to be tested; I know what awaits me in the case of a positive test, and I don’t want to wrap my life in the gloom of Huntington’s disease.*” This questionnaire could also see the burden of the COVID-19 pandemic and how it affects families“*. My husband and I are waiting for test results. Although I went to the tests with him, the doctor did not let me into the office or tell me anything—allegedly for the corona reason. At the same time, she was not wearing a face mask. I was sorry because it is difficult for both my husband and me*”.

It is important to understand why the persons decide to take a test and disclose their genetic status; we also respond to individuals who have witnessed HD in their relatives since childhood/adolescence. “*When I discovered that there was a 50% chance I could inherit this stuttering gene, I was determined to know what I was up to. Everything stopped making sense, and my only goal was to find out if I was sick or not*”. “*So that I wouldn’t ruin my partner’s life … And I would be ready for the institution and that I would be insane*”. “*To stop it in our family*”.

### 3.3. The Impact of HD

We divide this section into three subcategories to better understand the burden for caregivers, the effect on relationships in general, and the impact on families’ financial situation with HD.

#### 3.3.1. Burden for Caregivers

The results present the high burden of caregivers, showing higher burdens when coping with the mental symptoms rather than the motoric ones of their affected family members; “*My father and sister were HD patients. I saw aggression, the desire to kill, they ruined my life, nobody will bring me back my youth ages*”. It also affects the partnership “*The hardest thought is that my beloved husband, who is very wise, rational, just, will become someone who cannot take care of himself and lose his identity*”. The partners and caregivers at the same time are often overwhelmed with the care of the affected person in the family, and sometimes the care of the rest family comes short: “*I had a mother-in-law who was affected, and same time I have a husband and a son. I was caring for all; I would welcome any help*”.

#### 3.3.2. Relationships

The relationships are mostly affected within the family and involve different levels. The communication between parents and children can be damaged at different levels, as communication about HD implies a lot of challenges. Some respondents stated: “*My parents didn’t tell me anything about HD, although they knew about it. I blame them*”. The affected members experience problems communicating their genetic status in their own family and experience the feeling of guilt, “*Knowing about the HD is good when young people plan to start a family; I went to test myself for the kids, and now I don’t know how to tell them, that I am affected, I feel guilty*”.

The own perspective of the positive genetic result plays a role and reflects further in the relationship: “*The positive result destroyed me, my partnership; I don’t want to do it to them either*”. *some even stated*, “*everything about this disease puts me in depression so far … I try not to think about it and live. Still, every encounter with such a person takes away my zest for life*”.

#### 3.3.3. Impact on the Financial Situation of Families with HD

The impact on families’ financial situation with HD is very important, as most respondents describe it as unsatisfactory. It is very difficult as the partner or family needs to compensate for an affected member, “*I had to work full-time at retirement age, to be able to pay for my husband’s private caring facility*”. Most caregivers experience a difficult time as they have a full-time job and care for an affected individual and the rest f family. “*My husband was left without a job; the disability pension is so small. I have to go to work, and we don’t know what will happen next*”. The affected families need to rely on the help of further family members and wider relatives“*. We wouldn’t have been able to manage it without the help of our relatives*”. The situation is even more complicated for the early-manifest respondents who experience the problems due to the symptoms of HD: “*I haven’t been able to find a job for a long time, I act like I’m an alcoholic with my involuntary movements. And once I got a job, I almost everywhere had an accident at work. Currently, I am at the unemployment office, and I receive benefits, which, of course, are not even enough to cover the basic living costs like household and food; my family helps me*”.

### 3.4. Social and Health Services in the Slovak Republic

The emerging fact is the lack of support for social and health services in the Slovak Republic. Most respondents consider them to be insufficient, with the most common blame regarding inadequate knowledge about HD among the health professionals and staff working in the social services and nursing facilities. The respondents summarise their perspectives in the following strong statements: 

“*The patient is just waiting to die*”.

“*No support in the system*”.

“*Psychiatrists know nothing about it*”.

“*Inadequate, no one knows what to do*”.

“*Doctors and staff have minimal knowledge. Rather they take the disease as a sensation*”.

The participants were asked what could improve the health care and social systems. The majority of participants proposed rational and possibly easy solutions to enable better maintenance of affected individuals, such as *Better prescription of medical devices, prioritisation during the doctor visits, advice on how to deal with the dentist; insurance companies should also pay for the recreational stays for caregivers as well (not only affected person)*. It became clear that all respondents wish for more information and education about HD. 

## 4. Discussion

This is the first survey of its kind to examine the needs and problems of the Huntington’s landscape in the underserved region of Slovakia. Our survey results on the psychosocial impact of Huntington´s suggest similar conclusions to other studies in this field [18,31,32]. For overview purposes, we will discuss the different themes, which were clustered into subthemes based on responses.

### 4.1. Genetic Counselling and Testing

This survey highlighted the importance of proper guidelines for genetic testing and counselling [21], as implemented in many European countries. Physicians conducting the genetic test need to provide information about HD in a timely, private, and sensitive manner to those who wish to do so while respecting the interests of those who do not want the test [33]. The liberty of choice for the patient and the possibility to opt-out of the procedure at any step and re-enter the protocol if the patients wish so at the later stage of life [22,34]. The respondents at risk were determined not to undergo predictive genetic testing. They are afraid to receive irreversible information for the future and are worried about possible stigma [35]. However, higher importance should be given to genetic counselling on reproductive options as the option is regularly available in Slovakia. Consultation on reproductive options should be public for all interested persons, regardless of whether they have previously had to undergo predictive testing, or it should usefully be included in consultations on genetic testing if the person so wishes [20,36]. 

In addition, it should not be forgotten that individuals who learn that they will develop HD in the future needs to be offered support not only in a short time post-testing but even from the long time perspective [35]. An earlier survey of attitudes to predictive testing suggested that 11–15% of people at risk would consider suicide if given an unfavourable test result [35]. In contrast, Quaid et al. report that only four (2.1%) of the 189 individuals in the US predictive test sample were hospitalised after receiving unfavourable results [37]. The study of Almquist et al. even described only 0.97% of all participants who experienced a catastrophic response to predictive testing [38]. In summary, all reported frequencies are still high and require attention. 

However, rather rare negative reactions and impacts after the predictive testing should be compared to thousands of individuals who have been relieved of the anxiety of not knowing whether HD will develop [34,39]. Furthermore, in preventing fatal reactions after predictive testing, it is necessary to emphasise the importance of ongoing counselling regardless of the test result, identifying problematic reconciliation techniques and active support in reconciliation strategies [21]. Despite all recommendations about the appropriate psychological interviews as a part of predictive genetic testing, only a few respondents stated that they had attended an interview with a psychologist during this process. 

Moreover, in our survey, we specifically process the responses of individuals who have witnessed HD in their relatives since early childhood and adolescence. 

### 4.2. Family Relationships

HD has a major impact on family systems; impairments shape caregiver roles in the affected family member and change in time upon disease progression [40]. Our results confirmed the earlier findings of the role shifts in many HD families, including parenting (children overtaking the role of a parent in a family partly) [40]. In almost all areas, teenagers growing up in a family with HD actively assist in complex nursing care for a parent with HD. The topics that concern them include responsibilities and early decision-making, personal exposure and burden due to individual risk for HD. [41]. Some of our respondents describe their early experience with HD as negative; *my father and sister were HD patients. I saw aggression and desire to kill; they ruined my life, and nobody will bring me back to my youth.* Our findings are in line with interviews conducted with 32 young people living in a family with HD, where the following main problems were mentioned: the possibility of starting a family as a distressing decision; feelings of loneliness in the family and among peers, the experience of the affected parent being unable to have more reasonable conversations, and the non-affected parent being mostly described as absent and unreachable because they are taking care of a partner and often has several jobs for financial reasons; family life is so difficult and marked by the conflicts [42]. 

Our results are in line with numerous other studies highlighting the most affected domains in the family system involving family functioning, emotions and reactions, social functioning, and state and social services [41,42,43,44,45].

The respondents experience negative and traumatic experiences (witnesses of domestic violence, aggression, suicide and suicide attempts), causing long-term impacts on their own lives [42]. 

The hereditary aspect is also important as some respondents were, for a long time, not aware of their own risk as proper communication did not take place in the family [46] as also stated in our interviews: “*my parents didn’t tell me anything about HD*”,“ *we did not talk about it*”, “*I went to test me for the kids, and now I don’t know how to tell them, that I am affected, I feel guilty*”. 

Overall, the respondents also reported feelings of being overburdened and feeling alone without any support and, at the same time, caring for multiple jobs and households. 

### 4.3. Future and Family Planning

Future and family planning is one of the most important topics and reasons for undergoing predictive testing in clinical practice [47]. There is significant distress already when considering the predictive testing, and multiple factors, such as parenteral experience with disease and risk perception, play a role in decision-making [48]. Decruyenaere et al. observed that more than the majority of HD expansion mutation carriers, after undergoing predictive genetic testing, had chosen to have children with a prenatal or preimplantation genetic diagnosis. About one in three (35%) decided to have no children anymore. A minority (7%) was undecided or had no children for other reasons [49]. The factors impacting the decision not to have children were the personal experience of growing up in a family with HD and the ethical issues related to prenatal and preimplantation genetic diagnoses [49]. The factors associated with the non-use of prenatal testing and the preimplantation genetic diagnosis were aversion to terminating the pregnancy and optimism associated with treatment options and ongoing clinical trials for HD [50]. As we have already assumed, several factors play a role in the decision-making process, such as gender, knowledge and ethical issues about PD and PGD, the availability of these methods, the strength of the desire to have children, and personal experiences with HD in the family [45,49,50]. The results illustrate the complexity of the decision-making process and the necessity of in-depth genetic counselling. A group of those who knew about the risk and decided to have children had their decision mostly based on the hope of a cure that would come in time to help their children if needed; a certain role played the “wish thinking”, in which they believe that their children will not develop HD [49]. Findings similar to our responses were presented in the group of those who had children before they knew about the risk. They reflected the lack of information about HD and inheritance patterns [51]. In the group of those who choose not to have children because of the risk, the main reason was their own experience, how they saw the decline and death of family members for HD in generations [51]. 

### 4.4. Social and Health Care System Support

Our findings also highlight the central role of the physician or general practitioner who might consider working with families on strategies to improve emotional engagement, communication, and problem-solving and ensure better care for affected members [52]. The findings of studies point out the importance of support needed for families with HD facing different problems. There is an urgent need to develop strategies to help and support genetic counsellors, healthcare providers, and school workers [53]. Our respondents reported major complaints about the problematic and lacking support for patients with HD in the presymptomatic and especially advanced stage of the disease and reflecting the need for a multidisciplinary approach—the key role in helping build and develop the infrastructure played here patients and lay organisations. There is an urgent need for an interdisciplinary approach to HD management to ensure adequate systematic care during all stages of the disease. It is important to provide appropriate care at all stages of Huntington’s disease. Starting in the period before predictive testing with psychological support, involving the general practitioner, the specialist and the family, to balance early between fatalism (“I already have HD, and there is nothing I can do about it”) and denial (“These symptoms are not HD” or “I have no symptoms”) [21,23]. Available supports should include educational materials, home health care or skilled nursing support, respite care, advocacy organisations, and local or online support groups. Educating patients and caregivers about the availability of these services and providing access to them is imperative for neurologists and other healthcare providers [54].

## 5. Limitations

Although there are so far no epidemiological studies available in Slovakia, the prevalence of the HD is expected to be as in other European regions 6–15: 100,000 [55,56], with an anticipated increase of another 17% by 2030 [57]. Based on this information by the population of five million inhabitants in Slovakia, we emphasise that there are more affected families and individuals that we could not reach with our survey and would need to be identified in future. 

The one limitation, especially from the perspective of the qualitative research approach, was the online form of data collection. We preferred the questionnaire form with the possibility of answers in their own words to an interview via phone call or zoom, mainly to preserve the anonymity of the respondents and obtain answers with a lower degree of self-censorship. However, we are aware that the optimal way to obtain information would be a personal interview with the trained local HD medical specialist, who already knows the respondent and has a therapeutic relationship with them. However, this was not possible due to the lack of HD specialists in Slovakia and non-existent HD centres where patients would naturally concentrate. The number and type of participants could also be significantly influenced by the necessity of sufficient cognitive ability of patients to use the Internet, which we tried to cover with the possibility that the family could fill in the questionnaire on behalf of the patient. Another source of influence on the number and type of respondents could be the lack of access to the Internet, which according to data from Eurostat for the year 2020 [58]—at the time of data collection, on the level of access of households to the Internet represented only 14% of households in Slovakia that did not have access to the Internet.

In addition, the COVID-19-related isolation periods and restrictive measures may influence some responses. It should also be noted that the survey took place between the first two COVID-19 pandemic waves, although the first wave in Slovakia was very mild, and the measures were considered mild compared to other European countries. All of the responses were received within four weeks of publication in early October 2020, so the survey was completed quickly.

## 6. Conclusions

In summary, Huntington´s disease requires a multidisciplinary approach due to the different needs of those affected at different stages, which is still scarce in Slovakia. In routine clinical practice, health professionals often assume that after informing the patient and his/her blood relatives about the possibility of diagnosis by genetic testing and the real limitations of currently available treatment, no other kind of support is possible than symptomatic control of motor and psychiatric symptoms. The results of our survey on the experiences and suggestions for improvement of the HD community in Slovakia point mainly to the lack of basic education of doctors and the explicit absence of information about the possibility of prenatal and preimplantation genetic testing. Strategies are needed to improve the knowledge of health professionals and to provide better counselling. Furthermore, information on ongoing clinical trials, as well as research progress, is seen as important. The results also show that comprehensive social and health services need to be provided for the later stages of the disease. Experts and multidisciplinary teams should be formed, and public awareness of the disease should be raised to improve the situation in any underserved region.

## Figures and Tables

**Figure 1 jpm-12-01941-f001:**
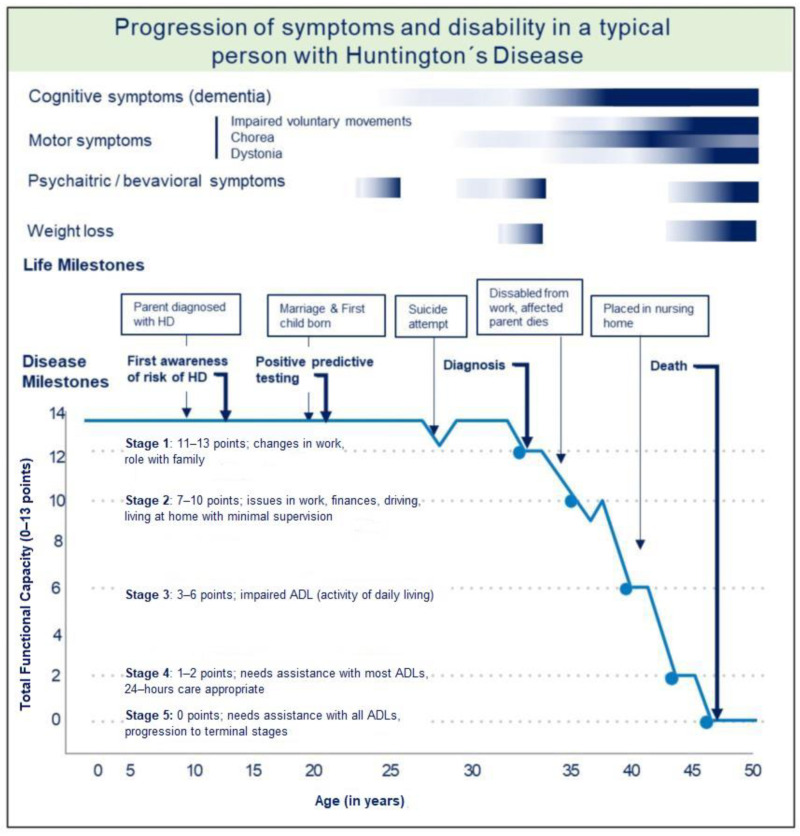
Progression of symptoms and disability in a typical person with HD, adapted from A Physician Guide to Management of HD [15] and onset of cognitive impairment adjusted by Langley et al., 2021 [16].

**Table 1 jpm-12-01941-t001:** The process of genetic counselling and testing for HD, adapted from [21]. Abbreviations: CAG—cytosine, adenine, guanine.

The Process of Genetic Counselling and Testing for HD	
Initial session	Providing details of family history to the doctor or genetic counsellor at the session, who then attempts to confirm the diagnosis history A doctor or genetic counsellor provides information about Huntington’s disease, the genetic test process and possible results Discussion on reasons for requesting a test at this stage and their approach to the possible outcomes. There will also be a chance to discuss, if wanted, the reproductive options available if the person wants to avoid the possibility of passing the disease on to the next generation Allowing identifying someone who will support them through the process Encourages them to consider the impact of any result on life, family or friends. This can be particularly important concerning family members who may not wish to be tested, but the test could also reveal their status Encouraging to consider financial implications and other issues such as life insurance and employment
Reflection period	The initial session is followed by a summary letter and a reflection period If the person wants to continue with the process, a second session is arranged
Second/third session	A review of the information discussed at the first session is undertaken. The doctor/genetic counsellor, by genetic testing guidelines, may consider a neurological and psychologicaö examination to evaluate the person. However, it is not a requirement for participation in predictive testing. Discussion about preparing for genetic tests and results
Genetic test	The date is arranged for a blood sample to be taken The results will not be provided on that day
The results	At a follow-up appointment (usually a month after), there will be a face-to-face discussion concerning the test results.
Follow-up sessions	Relevant follow-up sessions are arranged as required after results are given
Advice and support	Regardless of the result, the person must have access to support and advice available. It is appropriated to referral to local associations for HD, thus ensuring contact with others who are going through the same situation; there are mostly support groups and online communities

**Table 2 jpm-12-01941-t002:** Questions and answers of the respondents on the questionnaire administered via the Slovak HD association to patients, persons at risk, their relatives and partners, including their experiences and suggestions for improving the situation for the HD community in SR. The questions have been administered as multiple-choice or as open questions. The table shows an overview and a summary of the available answers. Abbreviations: GT—Genetic Testing; HD—Huntington Disease; N- number of participants; n.n. not known. PGD—Preimplantation genetic diagnosis; SSE—social services establishment; SR—Slovak Republic.

Questions in Individual Categories of the Questionnaire	Answers and Particular Groups of Respondents (Number of Participants)		
	Patients with HD (N = 5)/Asymptomatic HD Carriers (N = 4)	Individuals at Risk (N = 5)	Partners of Asymptomatic Persons (N = 6)/Partners of Manifest HD Patients (N = 10)
Knowledge about HD			
**Initial information** How and when did you first learn about the risk or presence of HD in your family or partner?	At the age: 14, 38, 42, 39, 50/15, 35, 50, n.n. From family: 6, From doctor: 3	At the age: 0, 14, 20, 38, n.n.	At the age: 7, 15, 25, 34, 39, n.n/25, 26, 30, 35, 38, 40, 46, 58, n.n, n.n From family: 5/1, From doctor: 1/5, From partner:0/4
**The main source of information** Where did you mainly get information about HD? How did your doctor teach you about HD? (open question)	From the family: 3; From doctor: 1, Internet: 7	From the family: 2, From doctor: 0, Internet: 3	From family:1, From doctor: 1, internet: 16
**Reproductive options** Has a doctor provided information on reproductive options, including PGD (your child would be healthy even if you/your partner is affected)?	Yes: 2 No, I am not interested anyway: 2 No, I searched by myself: 2 No, first time to hear: 3	Yes: 2 No, I am not interested anyway: 2 No, I searched by myself: 1 No, first time to hear: 0	Yes: 2 No, I am not interested anyway: 2 No, I searched by myself: 7 No, first time to hear: 3
**Experimantal treatment** Did you receive information on ongoing clinical trials for HD? How do you get any updated information?	Yes: 1 (doctor), 3 (Internet), family (2) No: 2 I am interested /I am not interested in updates: 8/0	Yes: 4 (Internet) No: 1 I am interested /I am not interested in updates: 5/0	Yes: 1 (doctor), 14 (Internet) No: 1 I am interested /I am not interested in updates: 15/1
**Lack of knowledge** What information do you think a person at risk for HD should receive? What did you find out over time, and would you appreciate it if you knew it before? (open questions)	PGD: 3 Consulting HD expert: 1 Early knowledge of HD risk (early diagnosis of parent/myself/not hiding HD in the family): 3; I don’t know: 2	Reprod. options, PGD: 1 Better awareness among health professionals: 1 Entitlement to a disability pension: 1; I don’t know: 2	PGD: 3 Information on research: 4 More support for HD community: 2 Consulting HD expert: 1; I don’t know: 6
**Genetic testing (GT)**			
**Reasons for (not) being tested** Please write down why you decided/consider being tested, resp. not to be tested? (open question)	For GT: know the truth: 2 Emerging symptoms: 3 Disability pension: 1 Future planning: 3	For GT: because of the children: 1 Against GT: live a fulfilled life: 1 I don’t know: 2	For GT: know the truth: 3 Future planning: 7 Disability pension: 1 Emerging symptoms: 3; I don’t know: 2
**Psychologist Consultation** Have you talked to a psychologist during genetic testing? Was it/was it not beneficial for you?	Yes: 2 (beneficial), 1 (not helpful) No: 7	Yes: 1 (beneficial), 1 (not beneficial) No: 3	Yes: 3 (beneficial), 2 (not beneficial) No: 11
**Impact of HD**			
**Most aggravating symptom** Which symptoms of HD are/were the most burdensome for you? (open question)	Positive test: 2 Motor skills (chorea): 5 Psychiatric (nervousness, cognitive decline): 2	Motor skills (chorea): 2 Psychiatric (depression, cognitive decline, aggression): 4 I don’t know: 1	Positive test: 1 Motor skills (chorea): 3 Psychiatric (cognitive decline, aggression): 7 Dysphagia: 3 Independence loss: 4
**Financial burden** How has HD affected your family budget? Was the help of the state sufficient for you? (open question)	Higher financial burden: 7 State support limited: 4, we are modest: 2, family help: 2. I don’t know: 2	Higher financial burden: 4 State support limited: we are modest 2, family help: 1 Any support yet: 1 I don’t know: 1	Higher financial burden: 13 State support limited: we are modest 3, family help: 5, extra job:1 Any support yet: 3 I don’t know: 3
**Impact on relationships** (answers resulted from other open-ended questions)	Blaming parents for hiding HD: 1 Fear of transmitting HD: 1 Hesitation/anxiety to say a positive result to the family: 2 Suspicion of alcoholism due to chorea: 2 Criminal prosecution: 2	Decision with the partner to ignore the fears of the HD and live the life that will come: 1	Anxiety about the gradual loss of a partner: 1 Fear of destroying the life of my healthy partner: 1 Fear of divorce: 2 Feeling my life is ruined due to illness of relatives: 1
**Social and health care services in SR**			
**Experiences** How do you evaluate social and nursing services for patients with HD in SR?	Insufficient—SSE staff and paramedics do not know HD: 5 I don’t know: 4	Insufficient—SSE staff and paramedics do not know HD: 2 I don’t know: 3	Insufficient—SSE staff and paramedics do not know HD: 9 Sufficient: 1; I don’t know: 6
**Suggestions for improvement** What do you think should be improved in social/health services for patients with HD? (open questions)	Better awareness among health professionals: 3 Support home care of HD patients: 1 Experts/Psychologists for HD: 1 I don’t know: 4	Awareness among health professionals and the public: 2 I don’t know: 3	Aids for later stages: 2 Special SSE for HD: 3 Better awareness: 5 Preferential appointment: 2 Support groups for both carriers and patients: Acceptance of the HD in the context of disability 1

## Data Availability

The data presented in this study are available upon request from the corresponding author. The data are not publicly available because the database contains patient personal data.

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
