# Peer review of "Psychosocial Impact of Huntington’s Disease and Incentives to Improve Care for Affected Families in the Underserved Region of the Slovak Republic"

_jpm, 2022, doi:10.3390/jpm12121941_

Round 1
Reviewer 1 Report (New Reviewer)
The article is interesting and generally well written. The language is appropriate. Some minor revisions are needed.
Abstract:
They should add the section of “Limitations” in the abstract.
Methods:
-
Line 177: “But with the possibility to answer in the text” delete the word “but”
-
Line 183: “The questionnaire was placed online” “Was available online”
-
Line 185/186: "...and the psychologist and was reviewed to ensure proper understanding and clear..."
Results:
-
Line 26: “Necessity” instead of “need”
Author Response
We would like to thank you and appreciate the time and effort that you have dedicated to providing your valuable feedback on our manuscript. We have been able to incorporate changes to reflect all of your suggestions provided. We have highlighted the changes within the manuscript. Here is a point-by-point response to comments and concerns.
Comment 1: Some minor revisions are needed. Abstract: They should add the section of “Limitations” in the abstract.
Response: We agree with this and have incorporated your suggestion in abstract.
Comment 2: Methods:
Line 177: “But with the possibility to answer in the text” à delete the word “but”
Line 183: “The questionnaire was placed online” à “Was available online”
Line 185/186: "...and the psychologist and was reviewed to ensure proper understanding and clear..."
Response: Agree. We have, accordingly, modified your coorection in abstract.
Comment 3: Results: Line 26: “Necessity” instead of “need”
Response: Agree, we have changed it.
Reviewer 2 Report (New Reviewer)
The authors report a qualitative study, with data collected from an online questionnaire rather than face to face interviews. Topics covered include the availability and rigor of the counselling process, the ramifications of genetic testing and the resources available through the health and social care system in Slovakia. Data were analysed using grounded theory. The authors found limitations in the counselling process, and participants highlighted the dearth of resources in managing their symptoms (or relative’s symptoms).
Comments
The study is well-described, and the analyses appropriately conducted. The conclusions drawn from the data are appropriate. The findings are, however of most interest to healthcare providers and public policy bodies in Slovakia.
I am not clear from the manuscript how they advertised the study beyond local HDA members – as omitting patients from outside this group is a potential source of bias.
Was the study online only? Again if study access was limited to internet users and patients with sufficient cognitive ability to use the internet this could potentially introduce bias by excluding important study populations.
Whilst questionnaires are useful to screen participants, ideally the participants would be intereviewed formally either by telephone/Zoom etc or in person, otherwise the data collected for qualitative analysis are quite limited.
I am not sure what the word ‘Tramandenously’ means - the authors should clarify this.
Author Response
We would like to thank you and appreciate the time and effort that you have dedicated to providing your valuable feedback on our manuscript. We have been able to incorporate changes to reflect most of your suggestions provided. We have highlighted the changes within the manuscript. Here is a point-by-point response to comments and concerns.
Comment 1: The study is well-described, and the analyses appropriately conducted. The conclusions drawn from the data are appropriate. The findings are, however of most interest to healthcare providers and public policy bodies in Slovakia. I am not clear from the manuscript how they advertised the study beyond local HDA members – as omitting patients from outside this group is a potential source of bias.
Response: Thank you for pointing this out. Therefore, we have added a clarification of the methodology in the "study assessment" section, as the Slovak HD patient society is not local but nationwide, and when it was renewed, there was an effort to reach out to future members in all regions of the Slovak Republic.
Comment 2: Was the study online only? Again if study access was limited to internet users and patients with sufficient cognitive ability to use the internet this could potentially introduce bias by excluding important study populations.
Response: Agree. Therefore, we have expanded section „study participants“ and „limitation“ to capture this issue.
Comment 3: Whilst questionnaires are useful to screen participants, ideally the participants would be intereviewed formally either by telephone/Zoom etc or in person, otherwise the data collected for qualitative analysis are quite limited.
Response: We agree with this and we have supplemented and expanded section „limitations“ to emphasize this point.
Comment 4: I am not sure what the word ‘Tramandenously’ means - the authors should clarify this.
Response: We have corrected it.
We look forward to hearing from you in due time regarding our submission and to respond to any further questions and comments you may have.
This manuscript is a resubmission of an earlier submission. The following is a list of the peer review reports and author responses from that submission.
Round 1
Reviewer 1 Report
The paper focus should be "Psychosocial impact of Huntington’s disease and incentives to improve care for affected families in an underserved region of the Slovak republic", yet the manuscript does not provide any insight on the peculiar issues related to HD families living in Slovak Republic. Both introduction and discussion are dominantly generic and referring to issues common to HD families worldwide.
Introduction is long and dispersive, describing in detail aspects only marginally related to the main topic of the paper (i.e. genetic deepening, Table 1 looks unnecessary). Bibliography is dated, references not always look appropriate and miss relevant content.
Figure 1 doesn't seem fit, as recent findings highlight as cognitive symptoms often develop years before the appearence of motor symptoms (Langley 2021, JNNP).
The methodology description does not detail the clustering process, nor how many reviewers were involved and what process was followed for the thematic analysis. Is is not specified which is the underserved Slovak region under analysis. The survey was realized in late 2020, it is likely that interviewees'opinions might have been biased by Covid19 spread and possible lockdown/isolation periods. The qualitative questionnaire was filled by a limited number of individuals
Discussion is too long and redundant, not centered on the main topic of the paper. Conclusions are generic and not relevant to this specific population.
Writing needs editing of typos and improving sentence clarity, tables look chaotic.
Author Response
We would like to thank you and appreciate the time and effort that you have dedicated to providing your valuable feedback on our manuscript. We have been able to incorporate changes to reflect most of your suggestions provided. We have highlighted the changes within the manuscript. Here is a point-by-point response to comments and concerns.
Comment 1: The paper focus should be "Psychosocial impact of Huntington’s disease and incentives to improve care for affected families in an underserved region of the Slovak republic", yet the manuscript does not provide any insight on the peculiar issues related to HD families living in Slovak Republic. Both introduction and discussion are dominantly generic and referring to issues common to HD families worldwide.
Response: We agree with this and have incorporated your suggestion throughout the manuscript mainly in introduction.
Comment 2: Introduction is long and dispersive, describing in detail aspects only marginally related to the main topic of the paper (i.e. genetic deepening, Table 1 looks unnecessary). Bibliography is dated, references not always look appropriate and miss relevant content.
Response: Agree. We have, accordingly, modified introduction, deleted table 1 to emphasize this point.
Comment 3: Figure 1 doesn't seem fit, as recent findings highlight as cognitive symptoms often develop years before the appearence of motor symptoms (Langley 2021, JNNP).
Response: Thank you for pointing this out. Therefore, we have changed figure 1 with relevant citation.
Comment 4: The methodology description does not detail the clustering process, nor how many reviewers were involved and what process was followed for the thematic analysis. Is is not specified which is the underserved Slovak region under analysis. The survey was realized in late 2020, it is likely that interviewees'opinions might have been biased by Covid19 spread and possible lockdown/isolation periods. The qualitative questionnaire was filled by a limited number of individuals
Response: We have changed methodology description and we added into limitation section influence of Covid-19 related isolation/lockdown.
Comment 5: Discussion is too long and redundant, not centered on the main topic of the paper. Conclusions are generic and not relevant to this specific population.
Response: Thank you for this feedback. We have shortened discussion and we have expanded conclusions. However we believe that detailed desription of mentioned studies in discussion could facilitate better understanding for other medical professionals who begins to be oriented in HD issue in any undersurved region.
Comment 6: Writing needs editing of typos and improving sentence clarity, tables look chaotic.
Response: Revised version has been edited by a native English speaker colleague. Tables was modified.
We look forward to hearing from you in due time regarding our submission and to respond to any further questions and comments you may have.
Reviewer 2 Report
The topic discussed by Hubcikova et al is very interesting and challenging and in my opinion, these studies are very much needed. It reflects well how not only the public and the society but health care professionals do not know much about this devastating disease. Before acceptance, I would like to suggest a very thorough editing process. In my opinion, the responsibility of medical doctors, social- and healthcare professionals should be more emphasized on how to educate patients and relatives about the progression of HD and how to cope with such a burden.
Author Response
We would like to thank you and appreciate the time and effort that you have dedicated to providing your valuable feedback on our manuscript. We have highlighted the changes within the manuscript. Here is a point-by-point response to comments and concerns.
Comment 1: The topic discussed by Hubcikova et al is very interesting and challenging and in my opinion, these studies are very much needed. It reflects well how not only the public and the society but health care professionals do not know much about this devastating disease. Before acceptance, I would like to suggest a very thorough editing process.
Response: Revised version has been edited by a native English speaker colleague.
Comment 2: In my opinion, the responsibility of medical doctors, social- and healthcare professionals should be more emphasized on how to educate patients and relatives about the progression of HD and how to cope with such a burden.
Response: Thank you for pointing this out. Therefore, we have incorporated your suggestion mainly in discussion – part „social and health care support“.
We look forward to hearing from you in due time regarding our submission and to respond to any further questions and comments you may have.
Round 2
Reviewer 1 Report
Authors have thoroughly and satisfactorily revised the paper.
For my understanding the paper refers to all Slovak Republic, so the title should in my opinion read "Psychosocial impact of Huntington’s disease and incentives to improve care for affected families in the underserved region of the Slovak republic".
Author Response
We would like to thank you again for your constructive feedback, we have incorporated your suggestion and highlighted it in the manuscript.
Comment : For my understanding the paper refers to all Slovak Republic, so the title should in my opinion read "Psychosocial impact of Huntington’s disease and incentives to improve care for affected families in the underserved region of the Slovak republic".
Response: We totally agree with this and have incorporated your suggestion.
We look forward to hearing from you in due time regarding our submission and to respond to any further questions and comments you may have.